# *Cryptosporidium varanii* Infection in Captive Leopard Gecko (*Eublepharis macularius*) and Its Association with Wasting Syndrome in Thailand

**DOI:** 10.3390/ani16010033

**Published:** 2025-12-22

**Authors:** Panasaya Nipithakul, Sasiwimon Yodpunya, Wareerat Prasitwiset, Nithidol Buranapim, Sahatchai Tangtrongsup, Saruda Tiwananthagorn

**Affiliations:** 1Faculty of Veterinary Medicine, Chiang Mai University, Chiang Mai 50100, Thailand; panasaya_seine@hotmail.com (P.N.); tangkwa-tang@hotmail.com (S.Y.); wareerat_pr@cmu.ac.th (W.P.); sahatchai.t@cmu.ac.th (S.T.); 2Nana Animal Hospital, Chiang Mai 50230, Thailand; armvet64@gmail.com; 3Research Center of Producing and Development of Products and Innovations for Animal Health and Production, Chiang Mai University, Chiang Mai 50100, Thailand

**Keywords:** *Cryptosporidium varanii*, wasting syndrome, leopard gecko, Thailand

## Abstract

Cryptosporidiosis is a major cause of gastrointestinal illness and severe diarrhea in immunocompromised hosts, and *Cryptosporidium* has also been widely reported in reptiles, where it can cause wasting syndrome and death. This study investigated whether *Cryptosporidium* infection contributes to the development of wasting syndrome in captive leopard geckos from a commercial breeding facility in Chiang Mai, Thailand. We monitored 35 captive leopard geckos across 23 enclosures over several weeks for four months. Each month, we collected fecal samples from their enclosures and examined them using microscopy and Polymerase Chain Reaction (PCR) to detect *Cryptosporidium*. Body weight, tail diameter, and body condition score were recorded to evaluate wasting syndrome. Slightly more than half of the geckos and their enclosures were infected with *Cryptosporidium*, while one in five individuals presented with wasting syndrome. *Cryptosporidium varanii* was identified by nested PCR targeting the 18S rRNA gene, followed by restriction fragment length polymorphism analysis and sequencing, which showed 100% similarity to reference *C. varanii* sequences in GenBank. *Cryptosporidium* infection was significantly associated with wasting syndrome, and persistent oocyst shedding was observed. This study provides the first evidence of *C. varanii* infection in leopard geckos in Thailand and underscores the need for greater awareness and expanded surveillance in Thailand and across the Asia-Pacific region.

## 1. Introduction

The popularity of reptile pets is increasing in Thailand. Leopard geckos (*Eublepharis macularius*) are the second of the most common species of reptiles presented in veterinary practices [1]. One of the common health problems that owners, breeders, and veterinarians often encounter is known as wasting syndrome or “going light”. This syndrome is characterized by anorexia, chronic weight loss, extreme emaciation (especially at the tail), dehydration, diarrhea, abdominal swelling, and death [2]. In many commercial reptile breeding facilities, this condition caused significant economic losses [3]. The etiology and pathogenesis of this syndrome are not clearly understood. Stress from poor management such as overcrowding, lack of heat, dehydration, irregular photoperiod, or inadequate diet may induce this syndrome [1]. Recently, several studies reported the association between wasting syndrome and cryptosporidiosis in leopard geckos [3,4,5].

Cryptosporidiosis is a disease caused by enteric protozoan parasites of the genus *Cryptosporidium*. It is considered one of the most significant diseases that cause gastrointestinal infection and severe diarrhea in immunocompromised humans and animals worldwide [6]. It has been identified in a wide range of vertebrate hosts, including mammals, birds, reptiles, and fishes [7]. The mode of transmission is the ingestion of *Cryptosporidium* oocysts via the fecal-oral route from one animal to another by direct or indirect contact, including food- and water-borne [8]. Two main *Cryptosporidium* species, *Cryptosporidium serpentis* and *Cryptosporidium varanii* (syn. *C. saurophilum*) have been isolated from the gastrointestinal tract of lizards [9]. In leopard geckos, *C. varanii* is more commonly associated with clinical disease and wasting syndrome, whereas *C. serpentis* has been predominantly linked to gastric disease in snakes and some other reptile species [3]. In addition, *C. parvum* was also reported in leopard geckos [3,10,11]. *C. varanii* causes a wide range of clinical signs, such as anorexia, progressive weight loss, dehydration, diarrhea, smelly feces, abdominal swelling, severe muscle wastage, and a high mortality rate of 50% [1,2].

Wasting syndrome in leopard geckos is multifactorial and not limited to cryptosporidiosis. Although intestinal *Cryptosporidium* spp. commonly causes malabsorption and progressive emaciation [4], inadequate nutrition is also a major contributor. Diets deficient in calcium, vitamin D, or fat-soluble vitamins can result in metabolic bone disease, poor growth, and systemic decline [12]. Poor husbandry (inappropriate temperature, humidity, UVB exposure, overcrowding, and sanitation) increases metabolic stress and susceptibility to disease [1,13]. Other causes include enteric parasitic coinfections (e.g., flagellates), fungal or bacterial infections, gastrointestinal obstruction, and neoplastic or systemic diseases, all of which can produce anorexia, catabolism, and weight loss [2,4,12].

Information on the occurrence of cryptosporidiosis and wasting syndrome in leopard geckos in Thailand has never been documented. It is necessary to determine whether *Cryptosporidium* infection is the actual cause of wasting syndrome to guide the appropriate treatment and control of these illnesses in the exotic pet industry. The objectives of this study were to estimate the infection rate of *Cryptosporidium* spp., the occurrence of wasting syndrome in leopard geckos in breeding farms in Chiang Mai, Thailand, and to determine the association between wasting syndrome and *Cryptosporidium* spp. infection in leopard geckos in a breeding farm in Chiang Mai, Thailand.

## 2. Materials and Methods

### 2.1. Animals

The leopard geckos in this study were obtained from a commercial breeding farm in Chiang Mai, Thailand. All geckos (35 geckos) were housed in translucent plastic containers (Figure 1), with 1–3 geckos per enclosure (23 enclosures). Each enclosure contained geckos of mixed sex (Figure 2). The environmental conditions were 25 °C with air conditioning during the breeding and laying season, and ambient temperature during the non-breeding season. Each box was provided clay bowl filled with a mixture of dried Sphagnum moss (*Sphagnum* spp.) and tap water; reptile sand was used for bedding. Ad libitum fresh water and mealworms (*Tenebrio* sp.) were provided in ceramic cups.

The use of animals in this study was approved by the Animal Care and Use Committee of the Faculty of Veterinary Medicine, Chiang Mai University (approval number R18/2564) in accordance with the Ethical Principles and Guidelines for the Use of Animals for Scientific Purposes, the National Research Council of Thailand.

### 2.2. Sample and Data Collection

Between May and September 2021, fecal samples were collected from 23 enclosures over three consecutive days and pooled into a single sample per enclosure. This pooled sampling procedure was repeated every 2 weeks for a total duration of 4 months. The pooled fecal samples from 23 enclosures were collected by S.T., P.N., and S.Y. and placed in a 5-mL tube. Each sample was labeled and stored at 4 °C prior to DNA extraction. The primary analytical unit for estimating infection prevalence was the enclosure-week, defined as the presence or absence of *Cryptosporidium* DNA in pooled fecal samples, as well as the enclosure status across the 4-month study period. Because some enclosures housed 2–3 geckos that were captured together, we assumed that any gecko in an enclosure with at least one positive pooled sample was classified as exposed to *Cryptosporidium* oocysts at that time point. To avoid overestimating individual-level prevalence, we conducted a sensitivity analysis limited to geckos that had individual confirmation by necropsy (if deceased) and nested PCR of tissue.

Animal information, including signalment (age and sex), body weight (g), body condition score (BCS), tail diameter (mm), and picture, was recorded for each leopard gecko every 2 weeks for 8 times, total period of 4 months, between May and September 2021 (Appendix A).

### 2.3. Determination of Wasting Syndrome

The BCS of each leopard gecko was determined using a previously described system [5]. The scoring system was defined (Figure 3) as one = emaciated (No fat deposits visible anywhere in the body, vertebral column easily visible, eyes appear sunken); two = thin (Minimal fat deposits in the body, the tail is straight and thin, the vertebral column is visible, but not prominent); three = average (Tail is slightly plump, the vertebral column is not easily seen, the abdomen is slightly rounded); Four = fat (Fat deposits have noticeably thickened the tail, the abdomen is wide in proportion to the body); five = obese (Tail is very rounded and thick; the abdomen is very wide in proportion to body size). Leopard geckos with a BSC of one or two were considered to have wasting syndrome.

BCS was assessed by a single veterinarian (N.B.) with expertise in exotic and wildlife medicine, and the assessment was blinded to both the enclosure information and the PCR results at the time of scoring. To ensure high intra-rater reliability, including consistent interpretation and application of the scoring standards, we established clear and explicit scoring criteria, developed a scoring manual with representative example images, and conducted two calibration rounds.

### 2.4. Determination of Cryptosporidium spp. Infections

#### 2.4.1. DNA Extraction

Fecal genomic DNA was extracted using NucleoSpin^®^ DNA Stool kit (Macherey-Nagel, Düren, Germany) following the manufacturer’s instruction. The genomic DNA (gDNA) was eluted in 50 µL of elution buffer, and the concentration and purity were measured using a spectrophotometer (Beckman Coulter DU^®^ 730 Life Sciences, San Jose, CA, USA) using wavelength at 260 and 280 nm. DNA samples were stored at −20°C until further analysis.

#### 2.4.2. Nested PCR—RFLP Analysis

A nested PCR assay targeting the fragment of the 18S rRNA gene was performed as previously described [14]. For the primary PCR step, a PCR product of about 1325 bp long was amplified by using primers 5′-TTCTAGAGCTAATACATGCG-3′ and 5′-CCCTAATCCTTCGAAACAGGA-3′. Briefly, the PCR amplification reactions were performed in a 20 µL reaction volume containing 20 ng gDNA (3–5 µL), 0.2 µM of each primer (0.4 µL of 10 µM), and 10 µL of 2× Quick Taq^®^ HS DyeMix (TOYOBO, Osaka, Japan). The DNA templates were subjected to 35 amplification cycles, each consisting of denaturation at 94 °C for 30 s, annealing at 55 °C for 30 s, and extension at 68 °C for 90 s, with a pre-denaturation at 94 °C for 2 min and a final extension at 68 °C for 7 min. PCR products were analyzed on a 1.5% agarose gel electrophoresis. For the secondary PCR step, a PCR product of about ~830 bp long was amplified from 2 μL of the primary PCR reaction mix by using primers 5′-GGAAGGGTTGTATTTATTAGATAAAG-3′ and 5′-AAGGAGTAAGGAACAACCTCCA-3′. The PCR and thermocycler conditions were identical to those used in the primary PCR, except that the extension time at 68 °C was reduced to 60 s. Distilled water (DW) and gDNA of *C. parvum* were used as negative and positive controls, respectively. In addition, DNA extraction blanks were periodically included to monitor contamination.

For species identification using RFLP, the secondary PCR products were single digested with 10 U of SspI (New England Biolabs, Ipswich, MA, USA) [14,15]. The restriction endonuclease SspI was employed for RFLP analysis, as this enzyme produces a characteristic restriction profile enabling discrimination of *C. varanii* from closely related *Cryptosporidium* spp., including *C. serpentis* and *C. parvum*, as reported previously [14,15]. The restriction reaction was carried out at 37 °C for 1 h, according to the manufacturer’s instruction. The digested PCR products were analyzed by 1.5% agarose gel electrophoresis and visualized after Redsafe™ (iNtRON, Boston, MA, USA) staining.

#### 2.4.3. DNA Sequencing and Phylogenetic Analysis

All positive *Cryptosporidium* spp. samples were submitted for capillary electrophoresis sequencing (CES) method (Macrogen, Sydney, Australia); however, sequencing data were successfully obtained for only 12 samples. The DNA sequences were compared with those in the GenBank database using the BLAST algorithm, and the species of *Cryptosporidium* present in the sample was determined. Nucleotide sequences were edited using BioEdit 7.1.3. Nucleotide sequences were aligned with the reference sequences from GenBank using ClustalX 2.1 software (http://www.ebi.ac.uk/tools/clustalw2, accessed on 5 November 2025). Phylogenetic relationships among the species were evaluated using the Maximum Likelihood method and the Kimura 2 parameter model of nucleotide substitutions, and the tree with the highest likelihood was selected using MEGA 12 [16,17], with branch reliability assessed by 1000 bootstrap replicates. A phylogenetic tree representing the *Cryptosporidium* spp. was then constructed. *Eimeria tenella* sequence (AF026388) was employed as the outgroup. All 12 representative amplicons were sequenced and deposited in GenBank under accession numbers [PX496096–PX496107]. The correspondence between sample ID, enclosure, and accession number is summarized in Appendix A.

### 2.5. Frequency and Persistence of Shedding

The 23 enclosures of leopard geckos were divided into 2 groups. One group (Negative group) consisted of boxes that tested negative at every time point, using nested PCR. The second group (Positive group) consisted of boxes that tested positive at any time using nested PCR. Fecal samples and data collection were collected as previously described (Sample and data collection).

### 2.6. Necropsy, Histopathology, and DMSO Modified Acid-Fast Staining Method

#### 2.6.1. Necropsy

At any time-point during this study, if an animal developed wasting syndrome and subsequently died, a complete necropsy was performed. During this study period, only one gecko died; it underwent a full necropsy with histopathological examination and *Cryptosporidium* identification in each organ using nPCR–RFLP. Visceral organs, including the stomach, small intestine, and large intestine, were removed and preserved in 10% neutral buffered formalin. Several representative sections from the stomach, small intestines, and large intestines were embedded in paraffin, sectioned at 5 μM, and stained with hematoxylin and eosin (H&E). Each section was examined by light microscopy, and the presence or absence of characteristic *Cryptosporidium* spp. organism and structure of each organ were noted.

#### 2.6.2. DMSO Modified Acid-Fast Staining Method and Microscopy

Intestinal contents from leopard geckos that underwent necropsy were examined. Each intestinal content was smeared onto a glass slide and stained using DMSO modified acid-fast technique as previously described [18]. The presence of *Cryptosporidium* spp. oocysts was determined under a light microscope by observing oocysts exhibiting morphological characteristics consistent with *Cryptosporidium* spp.

### 2.7. Statistical Analysis

The overall infection rate of *Cryptosporidium* spp. infection was calculated at both the cage level (23 enclosures) and the individual animal level (35 geckos). A sample was classified as positive if the fecal specimen tested positive at any sampling week using nested PCR. The overall proportion of wasting syndrome in the breeding farm was calculated as the percentage of geckos with a BCS ≤ 2 among all geckos examined (*n =* 35).

Descriptive statistics (mean ± standard error) were computed for biweekly measurements of body weight, tail diameter, and BCS. Longitudinal changes in these measurements between *Cryptosporidium*-positive and -negative groups across all time points were assessed using linear mixed-effects models, specifying animal identification (ID) as a random intercept and including infection status, week, and their interaction as fixed effects.

Associations between *Cryptosporidium* infection and potential explanatory variables (cage type, sex, and age) were evaluated using univariable logistic or exact logistic regression, depending on data distribution. Likewise, risk factors for wasting syndrome were examined using univariable and multivariable logistic regression models. Cage type and age category were retained as adjustment variables in all multivariable models regardless of statistical significance. Sex was invalid to be included in the multivariable model. Variables with *p* < 0.250 in univariable analyses were considered eligible for multivariable model building using a backward elimination procedure. A *p* value < 0.05 was considered statistically significant. All statistical analyses were performed using Stata version 16.1 (StataCorp, College Station, TX, USA).

## 3. Results

### 3.1. Overall Infection Rate and Persistence of Cryptosporidium Infection

From nested PCR, positive samples revealed a single ~830 bp amplicon. With RFLP analysis using SspI, the enzyme digested the single 18 rRNA PCR amplicon and revealed 5 bands, including 18, 33, 109, 255, and 418 bp. However, only three prominent amplicons of 109, 255, and 418 bp were visible on the agarose gel (Figure 4). Of the 23 enclosures, 12 boxes were detected as positive for *Cryptosporidium* spp. using nested PCR-RFLP, resulted in the overall infection rate of 52.17% (95% CI: 30.6–73.2) in this breeding farm. When considering infection individually, we found 51.43% of leopard geckos positive with *Cryptosporidium* spp. (18/35; 95% CI: 30.48–81.28).

Twelve positive samples were submitted for sequencing; all positive samples were identified as *Cryptosporidium varanii* (syn. *C. saurophilum*). Variations were observed among the 18S rRNA gene sequences of *C. varanii* isolates from leopard geckos, with a T-to-C single-nucleotide substitution (Appendix A). The sequence obtained for 5 isolates LG11, LG61, LG66, LG79, LG31 varied by a single base substitution (C/T). The ‘T’ cluster revealed 100.00% homology with *C. varanii* in leopard gecko in Spain (EU553551; [11]), and *C. varanii* in a pet snake in Thailand (KM870593; [19]), various captive snakes in China (MT626665; [20], PP101400; [21]). The ‘C’ cluster revealed 100.00% homology with *C. varanii* from leopard gecko in Spain (EU553556; [11]), and *C. varanii* in various captive snakes in China (PP101401, PP088095; [21]) (Figure 5). When compared with *C. serpentis*, BLAST analysis revealed 92.70–99.86% identity, supporting classification as *C. varanii.*

The frequency of shedding for each box during May–September 2021 is shown in Appendix A. Ten leopard geckos shed oocyst persistently (consecutively ≥2 positive of 4 time points), including No. 004–006, 009–010, 022, 027–028, 031–032 (Appendix A).

### 3.2. Occurrence of Wasting Syndrome in This Breeding Farm

Of the 35 leopard geckos in this breeding farm, 7 geckos experienced wasting syndrome as determined by the body condition score (BCS) system. The proportion of wasting syndrome in this breeding farm was 20.00% (95% CI: 8.4–36.9).

### 3.3. Gross and Histopathological Findings in Deceased Leopard Geckos

During the study period, one leopard gecko died and underwent necropsy and histopathological examination. The female leopard gecko (ID 029, Appendix A) was 1–5 years of age, carcass weight was about 17.2 g. Grossly, the body condition score was one; no fat deposits were visible anywhere in the body, the vertebral column was easily visible, the eyes appeared sunken, and the width of the tail base was markedly reduced (Figure 6a). Coelomic fat stores were absent. The small intestinal serosa was reddened, while the proximal part of large intestine was distended by yellowish content (Figure 6b).

From the intestinal contents smear, followed by staining using DMSO-modified acid-fast stain, revealed the presence of pink and round oocysts of 4–6 μM diameter on a pale green background (Figure 7), with the distinct oocyst wall compatible with *Cryptosporidium* spp.

Erosion of the intestinal epithelium in leopard geckos was observed (Figure 8), with variable numbers of *Cryptosporidium* spp. oocyst present with a patchy distribution among the intestinal epitheliums. *Cryptosporidium* spp. oocysts were recognized as round, pale, basophilic organisms 3–6 μM in diameter. In the stomach, we observed mild epithelial changes without apparent organisms. In addition, nested PCR-RFLP from the stomach, small intestine, and large intestine could confirm *Cryptosporidium* infection.

### 3.4. Association of Cryptosporidium spp. and Body Weight, Tail Diameter, and BCS

Body weight showed a trend toward association with *Cryptosporidium* infection (*p* = 0.07) (Table 1). Geckos that tested positive for *Cryptosporidium* weighed, on average, 8.93 g less than negative individuals at baseline. Body weight increased significantly over time, with geckos gaining approximately 0.90 g per week (95% CI: 0.21–1.60). However, the weekly weight-gain trajectories did not differ between *Cryptosporidium*-positive and -negative groups (Table 1, Figure 9).

Tail diameter was significantly associated with infection status (*p* = 0.003). At baseline (Week 1), *Cryptosporidium*-positive geckos had tail diameters that were 3.16 mm smaller than those of negative geckos. Tail diameter also increased steadily across weeks at a rate of 0.43 mm per week (95% CI: 0.23–0.63), and this growth rate was comparable between the two infection groups (Table 1, Figure 9).

BCS of *Cryptosporidium*-positive geckos was, on average, 0.41 points lower than that of negative individuals, although this difference did not reach statistical significance. No other variables demonstrated significant associations with BCS (Table 1, Figure 9).

#### 3.4.1. Association of *Cryptosporidium* spp. Infection with Sex and Age (Table 2)

Of 35 leopard geckos, 18 were kept individually, and 7 (38.89%) were *Cryptosporidium* positive, 47.88% of female geckos but all male (100%), and 50.00% of geckos less than 1 year, and 51.62% of geckos between 1–5 years were *Cryptosporidium* positive. From univariable logistic regression, there was no association between *Cryptosporidium* spp. infection with enclosure type, sex, and age (*p* ≥ 0.05).

#### 3.4.2. Association of Wasting Syndrome with *Cryptosporidium* spp. Infection, Cage Type, Sex, Age, Body Weight, Tail Diameter, and BCS (Table 3 and Table 4)

Among 18 *Cryptosporidium*-positive leopard geckos, 6 (33.33%) experienced wasting syndrome, while no gecko in the negative group experienced wasting syndrome. From univariable logistic regression, there was a trend toward the association between wasting syndrome with *Cryptosporidium* spp. infection (OR = 8.00, 95% CI = 0.847–75.56; *p* = 0.07) (Table 3).

Of the total 35 leopard geckos, 15.38% of geckos in individual cages, 18.18% of geckos in group cages, 21.88% of females, but no males, 25.00% of geckos age less than 1 year and 19.35% of geckos age between 1 and 5 years, experienced wasting syndrome during the study time frame. From univariable logistic regression, there was no association between wasting syndrome with sex, cage type, and age (*p* ≥ 0.05) (Table 3).

Nevertheless, the univariable logistic regression analysis revealed significant associations between wasting syndrome and body weight, tail diameter, and BCS. It was found that for every 1 g of body weight, the odds of wasting syndrome decrease by 11%; for every 1 mm increase in tail diameter, the odds of wasting syndrome decrease by 83%; and for every 1 score increase in BCS, the odds of wasting syndrome decrease by 96% (Table 3).

A multivariable logistic regression model was constructed to identify factors associated with wasting syndrome in leopard geckos and is shown in Table 4. Cage type, age category, *Cryptosporidium* infection status, and tail diameter were included in the final model. Cage type and age categories were retained in the model as an adjustment variable. Body weight was excluded due to collinearity with tail diameter and lack of biological interpretability in adjusted models. BCS was excluded due to a perfect outcome predictability.

Geckos housed in cages containing more than 1 individual had significantly higher odds of developing wasting syndrome (OR = 13.64, 95% CI: 1.28–145.78). Although geckos younger than 1 year tended to have a higher risk (OR = 7.14, 95% CI: 0.60–84.71), this association was not statistically significant (*p* = 0.119). *Cryptosporidium* infection was a strong independent risk factor, with infected geckos having over 11 times higher odds of developing wasting syndrome compared with uninfected geckos (OR = 11.15, 95% CI: 1.78–69.98). Tail diameter demonstrated a significant protective factor. Each 1 mm increase in tail diameter reduced the odds of wasting syndrome by approximately 67% (OR = 0.33, 95% CI: 0.21–0.50), highlighting the critical role of fat reserves in the development of wasting syndrome. Overall, the final multivariable model demonstrated excellent explanatory power (Pseudo R^2^ = 0.71), supporting the importance of *Cryptosporidium* infection and reduced tail fat reserves as key contributors to wasting syndrome in this population.

## 4. Discussion

*Cryptosporidium* infection has been described in many different reptile species, including leopard geckos [11,22]. The present study represents the first identification of *C. varanii* infection in leopard geckos in Thailand. The overall infection rate of *Cryptosporidium* spp. infection in this breeding farm was 51.43%, which is higher than previous reported rates of 16.0% (74/462) in Austria [11] and 21.9% (7/32) in Spain [10]. While the proportion of *C. varanii* infection in this breeding farm was 52.17%, much higher than previous studies: 6.9% (32/462) in Austria [11] and 9.4% (3/32) in Spain [10]. Previous studies have reported various *Cryptosporidium* species in leopard geckos, including *C. varanii, C. serpentis*, and *C. parvum* [3,10,11,23]. However, only *C. varanii* have been identified from the leopard geckos in this study. This is consistent with the previous finding in Thailand, where *C. varanii* (or *C. saurophilum*) was also the only species identified in corn snakes [24], suggesting a possible species predominance in captive reptiles in the region. The observation of 99.86% identity with *C. serpentis* likely reflects the conserved nature of the target 18s rRNA gene and highlights a known limitation of BLAST-based species assignment for *Cryptosporidium*. Consequently, species designation in this study was based primarily on phylogenetic inference, which demonstrated robust clustering with reference *C. varanii* sequences and separation from *C. serpentis*, consistent with previously published classifications.

Necropsy of leopard geckos that tested positive for *Cryptosporidium* spp. and died with clinical signs of wasting syndrome revealed pink, round oocysts (4–6 μM) in intestinal content smears, consistent with previous descriptions in this species [3]. Gross lesions resembled those reported by Terrell [5], including marked narrowing of the tail base, absence of coelomic fat, reddened small-intestinal serosa, and variable numbers of *Cryptosporidium* oocysts distributed along the intestinal epithelium. DNA sequencing of PCR-positive amplicons obtained from stomach, small intestine, and large intestine confirmed the presence of *C. varanii* in all sampled sites. Recent reports have likewise documented *Cryptosporidium* spp. in multiple gastrointestinal segments of leopard geckos and other lizard species. Histopathological observations in wild-caught spiny-tailed lizards include catarrhal enteritis with villous atrophy and muscular edema of the small intestine [25], whereas affected leopard geckos have shown mild gastric epithelial hyperplasia, mucosal thickening with lymphocytic infiltration, and enterocyte hyperplasia with villous thickening in the intestine [5]. Taken together, the findings from this study and prior literature indicate that *C. varanii* infection in reptiles may involve both the stomach and the intestinal tract, in contrast to the earlier assumption that this species is restricted to the intestine. Because the anatomical distribution of lesions appears to vary among reptile hosts and *Cryptosporidium* species, it may be more appropriate to describe *C. varanii* as capable of causing gastrointestinal involvement, rather than classifying it strictly as an intestinal parasite.

Body weight and tail diameter, the key indicators of body condition in leopard geckos, showed clear differences associated with *Cryptosporidium* infection. Although baseline body weight was not significantly different, infected geckos were nearly 9 g lighter, consistent with reports that cryptosporidiosis is associated with chronic weight loss in reptiles [26]. Biweekly weight-gain trajectories were similar between groups, suggesting that infected geckos retained compensatory growth capacity, during devasting in the breeding seasons, when provided with optimal husbandry, as also noted in managed reptile cohorts [23]. In addition, the 1st–4th-week period was a laying period, and geckos were kept at 25 °C with air-conditioning, while the 5th–8th-week period was a non-breeding period, and geckos were kept at an ambient temperature, which was the period during which reproductive stress may have decreased and increasing appetite occurred, resulting in weight gain and increase tail diameter and BCS.

Tail diameter showed a statistically significant association: infected geckos had markedly smaller baseline tail diameters, reflecting reduced fat reserves and early wasting [27]. Despite this, tail growth over time was comparable, underscoring the critical role of tail fat reserves as the primary energy store in geckos. Individuals with greater adipose reserves may be better able to withstand the metabolic consequences of chronic infection [27]. BCS was lower in infected geckos, but the difference was not statistically significant. Overall, infection appears to reduce initial body reserves, particularly tail fat, while growth potential under controlled conditions remains largely preserved. Although BCS tended to be lower in infected geckos, the difference did not reach statistical significance. This may reflect limited sample size or the relatively coarse resolution of ordinal scoring systems compared with quantitative morphometric indicators. Prior reptile studies have similarly noted that BCS scales may be less sensitive to early nutritional deterioration caused by enteric protozoal infections [4]. The final multivariable model in the present study demonstrated excellent explanatory capacity, reinforcing the combined importance of *Cryptosporidium* diminished fat stores in the development of wasting syndrome. These findings highlight the need for both infection control and routine monitoring of tail condition in captive populations.

The present study was the first study in Thailand to elucidate the association between wasting syndrome and *Cryptosporidium* spp. infection in leopard geckos. In univariable analysis, the variable displayed a near-significant association (*p* = 0.07). Although this exceeds the traditional 0.05 threshold, the observed trend is consistent with the hypothesized relationship and may reflect a true underlying effect that this study was underpowered to detect definitively. Other causes of wasting syndrome include stress from poor management, such as overcrowding, lack of heat, dehydration, irregular photoperiod, or poor diet, that may induce this syndrome [1]. In multivariable logistic regression analysis, nonetheless, *Cryptosporidium* infection was the major independent predictor of wasting syndrome, with infected individuals exhibiting more than elevenfold higher odds of developing the condition. This strong association is consistent with the pathogen’s ability to induce chronic enteropathy, reduce nutrient absorption, and cause progressive deterioration in body condition, corresponding with the previous studies reported by Terrell, Deming, and Dellarupe [3,4,5]. Although infection was statistically associated with wasting syndrome, the pooled sampling design and potential confounding by seasonal, reproductive, and temperature-related factors preclude causal inference. The observed association should therefore be interpreted as hypothesis-generating rather than definitive evidence of causation. These findings suggest a meaningful trend but also highlight the need for larger studies with individual-level sampling to refine effect estimates.

Although there was no study about the characteristics of fecal shedding of *C. varanii*, there were multiple reports about the characteristics of fecal shedding of *Cryptosporidium* spp. in other reptiles. Intermittent fecal shedding of a variable number of oocysts was reported in snakes [28,29,30]. The current study could elaborate that the affected geckos tend to shed oocysts in their feces intermittently. Ten geckos that were tested positive for *C. varanii* for at least 2 times from 4 times using nPCR demonstrated the persistent infection of *C. varanii.* However, these animals did not experience wasting syndrome and were generally healthy over the entire time. It seems that the presence of *C. varanii* is not always associated with clinical signs of cryptosporidiosis, including anorexia, progressive weight loss, dehydration, diarrhea, smelly feces, abdominal swelling, and severe muscle wastage [1,2]. These geckos may represent a group of geckos with subclinical infection or a carrier for *Cryptosporidium* spp. infection, and it is possible that geckos that experienced *Cryptosporidium* spp. infection with subclinical infection may clear the infection by themselves, which is similar to a previous study by Deming [4]. Therefore, repeated fecal testing and using more than one diagnostic method are required in suspected cases. Overall, our findings indicate that *Cryptosporidium* infection in leopard geckos is associated with reduced baseline body reserves, particularly tail fat, but growth potential under controlled environmental conditions may remain preserved. This pattern is consistent with the chronic, often subclinical course of *C. varanii* infection, in which clinical severity depends heavily on nutritional status, stress, and husbandry quality [27].

*Cryptosporidium* oocysts are very resistant to chemical disinfection. Several studies have reported the efficacy of many disinfectants on the infectivity of *Cryptosporidium* oocysts; chlorine and chloramine have very little to no effect [31,32,33]. Treatment with chlorine dioxide required high concentration and contact time for inactivation [34]. The 70% ethanol, 37% methanol, 6% sodium hypochlorite, and 70% isopropanol did not reduce the infectivity of *C. parvum* oocysts in cell culture [35]. On the other hand, one study reported that 6% Hydrogen peroxide could be effective against *Cryptosporidium*; it reduced the infectivity of *C. parvum* oocysts in cell culture [35]. Therefore, choosing effective disinfectants for equipment disinfection is important and helps reduce the transmission of *Cryptosporidium* spp. infection. Currently, no effective treatment has proven to eliminate cryptosporidiosis in leopard geckos, according to previous studies. Paromomycin sulfate has been reported to improve clinical signs; however, after treatment was stopped, shedding of oocysts and clinical signs recurred [36]. Treatment with azithromycin was not effective and did not show any effect on the oocyst shedding; nitazoxanide and rifaximin were also found to be not successful for treatment [1,37]. Hyperimmune bovine colostrum (HBC) has been used in moribund leopard geckos infected with *Cryptosporidium*. However, it took several weeks to be effective [38]. Therefore, the separation or euthanizing of infected animals is suggested, especially in severely emaciated geckos with a poor prognosis and a possible source of infection to other geckos.

This breeding farm first received leopard geckos from another farm in Bangkok, Thailand, which may have been the source of *Cryptosporidium* spp. infection in this farm. In the breeding season, male geckos were transferred from box to box for breeding. Female geckos laid eggs for several times. When the newborn geckos were born, they were put together in the same box with a clay bowl filled with a mixture of dried Sphagnum moss (*Sphagnum* spp.) and tap water, and reptile sand was used for bedding. Most of them were transported to other exotic pet shops, which could facilitate the spread of *Cryptosporidium* spp. oocyst to other areas, and the rest were kept for replacing breeders, which possibly became infected when they were introduced to the breeding group. Other possible causes for the spread of *Cryptosporidium* spp. oocyst in this farm was fomites, including equipment for feeding, cleaning the cage, and reptile sand for bedding. All equipment was reused and did not receive any disinfection. From the possibility of transmission by fomites, this might be an important reason for the risk association analysis that found no association between *Cryptosporidium* spp. infection and cage type. However, an individual box for each leopard gecko is still recommended to prevent infection via the fecal-oral route, along with quarantine and separate equipment for each box. From a practical standpoint, breeders should implement quarantine for newly introduced geckos, use enclosure-dedicated tools, and maintain strict hygiene protocols. Disinfectants with proven efficacy against *Cryptosporidium* oocysts, such as ammonia-based products, steam, and selected peroxide compounds, should be used as part of routine cleaning. Test-and-separate strategies, in which PCR-positive enclosures are segregated from negative ones and breeding stock are preferentially selected from consistently negative enclosures, may help reduce transmission within commercial facilities.

The limitations of this study were sample collection, as individual sample collection was not fully available, and a small sample size, as well as the number of animals with wasting syndrome was limited. Comprehensive surveillance of *C. varanii* infection and wasting syndrome, and risk analysis with a larger sample size, are suggested for further application in disease control in breeding farms of leopard geckos.

From a One Health perspective, intensive breeding of exotic reptiles may facilitate the maintenance and spread of *Cryptosporidium* spp. within captive populations and potentially into other animal collections. Although zoonotic transmission of *C. varanii* to humans has not been definitively demonstrated, close contact between breeders, veterinarians, and infected geckos underscores the importance of hygiene, personal protective equipment, and surveillance for *Cryptosporidium* spp. in both animal and human populations. Our findings thus contribute to a broader understanding of cryptosporidiosis at the animal–human interface.

## 5. Conclusions

This study provides new insights into the *C. varanii* infection in leopard geckos from a breeding farm in Chiang Mai, Thailand. It is the first to confirm that *Cryptosporidium* infection is a strong independent risk factor of developing wasting syndrome compared to uninfected geckos. In addition, the multivariable logistic regression demonstrated excellent explanatory power and supports the importance of *Cryptosporidium* infection and reduced tail fat reserves as key contributors to wasting syndrome in this population. Infection with *C. varanii* in breeding stock could be transmitted to other farms across the country. Awareness of this disease should be increased, and comprehensive surveillance across Thailand and the Asia-Pacific region is needed.

## Figures and Tables

**Figure 1 animals-16-00033-f001:**
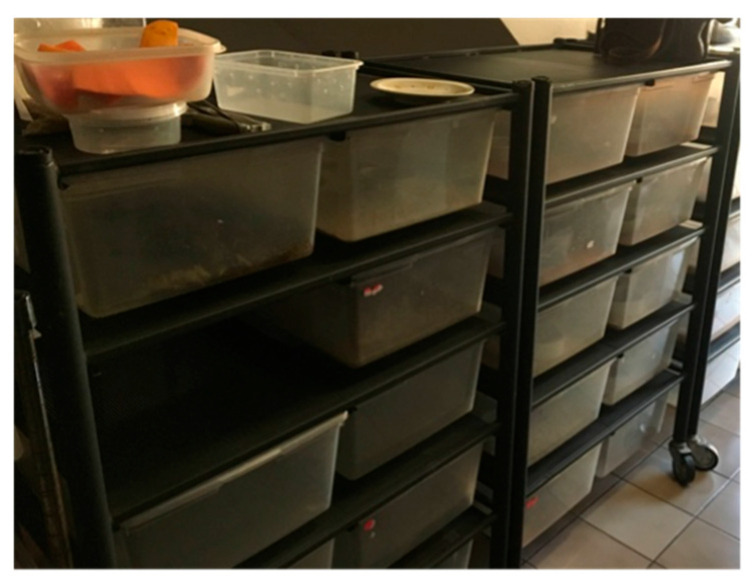
Farm environment and husbandry conditions of study farm.

**Figure 2 animals-16-00033-f002:**
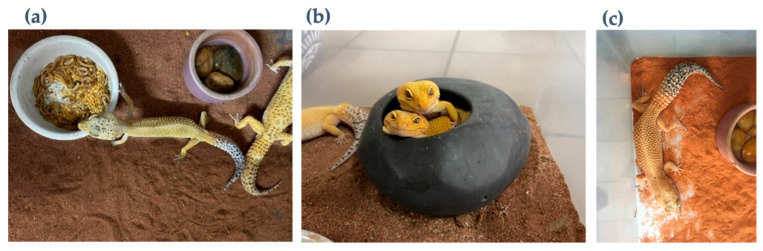
Breeding enclosures for group cage type (**a**), hiding box in group enclosure (**b**), and individual enclosure (**c**).

**Figure 3 animals-16-00033-f003:**
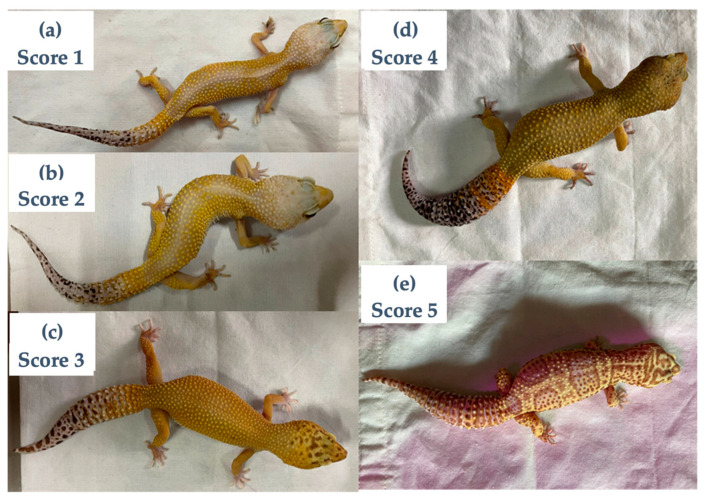
Leopard geckos with different body condition scores (BCS). (**a**–**e**): scores 1–5, respectively.

**Figure 4 animals-16-00033-f004:**
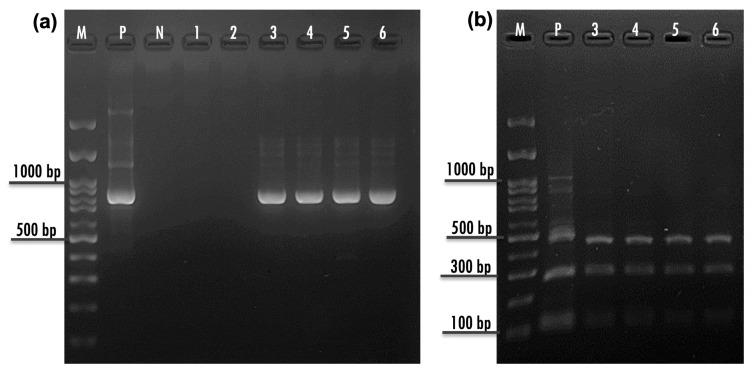
Detection of *Cryptosporidium* spp. infection in by nested PCR (**a**) and RFLP analysis patterns after SspI (NEB, Catalog # R0132S) digestion (109 bp, 255 bp, 418 bp) (**b**). M: 100 bp DNA ladder, P: Positive control (*C. parvum*), N: Negative control (DDW), 1–2: Negative samples, 3–6: Positive samples.

**Figure 5 animals-16-00033-f005:**
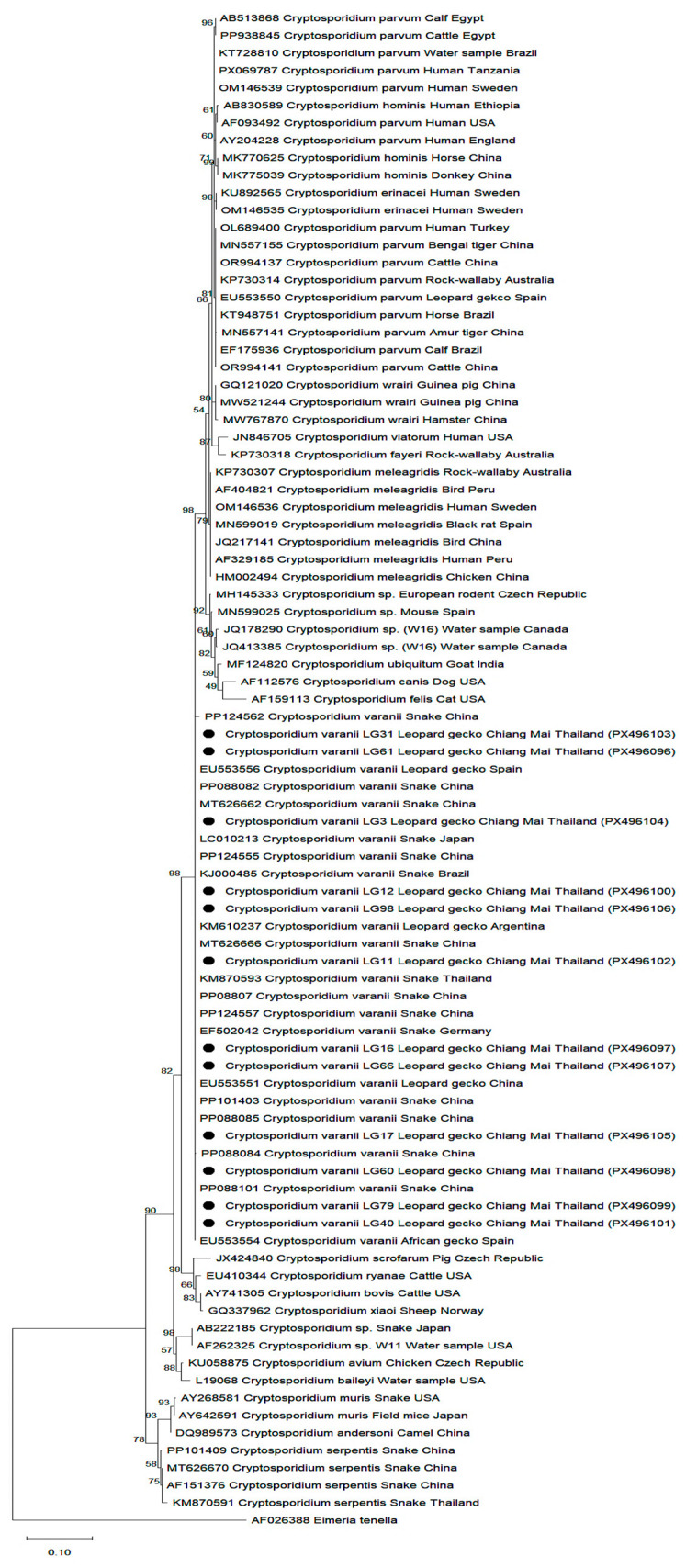
Phylogenetic relationships of *Cryptosporidium varanii* 18S rRNA gene in this study to various species of other known *Cryptosporidium* protozoa. The analysis was inferred using the Maximum Likelihood method and the Kimura2-parameter model of nucleotide substitutions, and the tree with the highest log likelihood (−2736.23) is shown. Sequences were retrieved from GenBank, aligned using ClustalW, and analyzed using MEGA 12 software. The analytical procedure encompassed 86 nucleotide sequences with 687 positions in the final dataset. Sequences of *Cryptosporidium varanii* found in this study are marked with a ●.

**Figure 6 animals-16-00033-f006:**
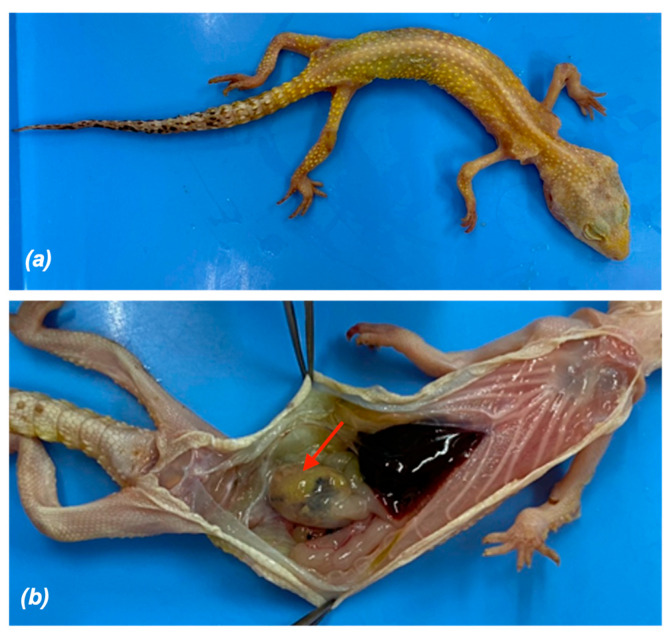
Necropsy and external appearance of carcass; (**a**) body condition score was one, no fat deposits visible anywhere in the body, vertebral column easily visible, eyes appear sunken, and the width of the tail base was markedly reduced. (**b**) The proximal part of the large intestine was distended by yellowish intestinal content (arrow).

**Figure 7 animals-16-00033-f007:**
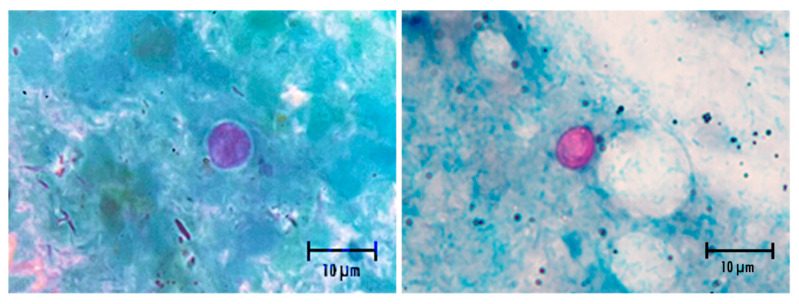
Intestinal contents smear: Round oocysts of 4–6 μM diameter on a pale green background compatible with *Cryptosporidium* spp.

**Figure 8 animals-16-00033-f008:**
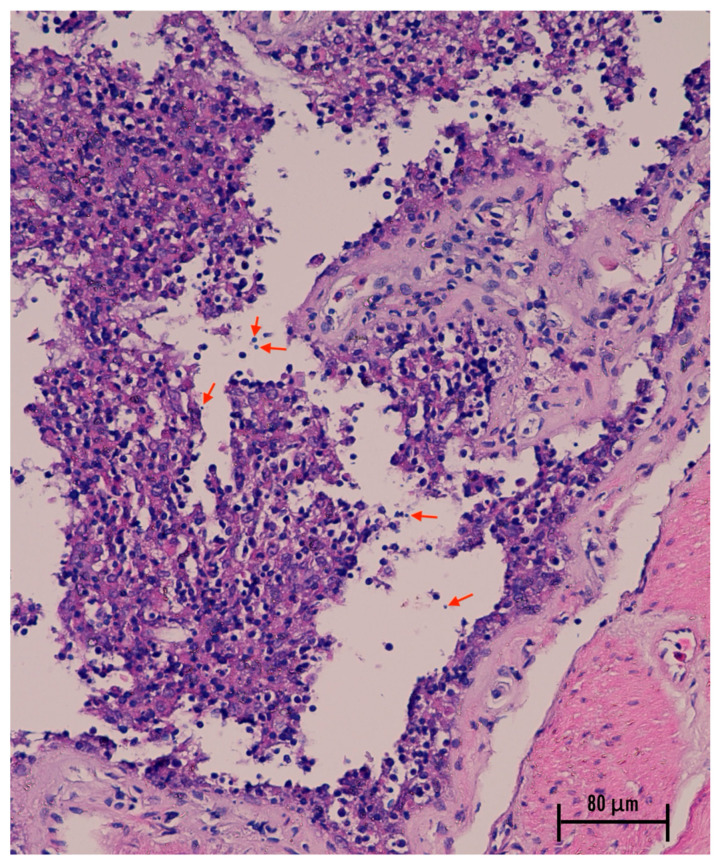
Histopathological lesion of the intestine of the leopard gecko with *Cryptosporidium* infection; arrows indicate suspected *Cryptosporidium* organisms.

**Figure 9 animals-16-00033-f009:**
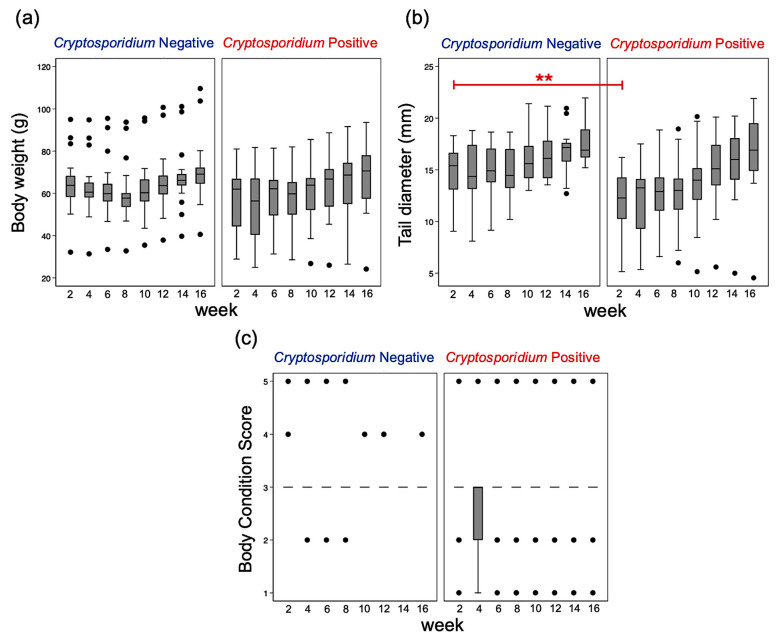
Boxplots showing biweekly body weight (**a**), tail diameter (**b**), and body condition score (**c**) in *Cryptosporidium*-negative and -positive leopard geckos. Boxes represent the interquartile range (IQR), horizontal lines indicate medians, whiskers denote the data range, and dots represent outliers (** *p* < 0.001).

**Table 1 animals-16-00033-t001:** Linear mixed-effects model results for body weight, tail diameter, and body condition score in *Cryptosporidium*-positive and negative leopard geckos.

Outcome	Effect	Estimate (β)	95% CI	*p*-Value
Body weight (g)	*Cryptosporidium* (positive vs. negative)	−8.93	−18.60–0.75	0.070
	Week (1–8)	0.90	0.21–1.60	0.010 *
	Crypto × Week	0.69	−0.29–1.67	0.166
Tail diameter (mm)	Cryptosporidium (positive vs. negative)	−3.16	(−5.26)–(−1.06)	0.003 *
	Week (1–8)	0.43	0.23–0.63	<0.001 *
	Crypto × Week	0.20	−0.09–0.48	0.177
BCS	Cryptosporidium (positive vs. negative)	−0.41	−0.90–0.07	0.095
	Week (1–8)	−0.01	−0.05–0.02	0.533
	Crypto × Week	0.02	−0.03–0.07	0.417

* Statistically significant difference.

**Table 2 animals-16-00033-t002:** Univariable logistic regression analysis of variable associated with *Cryptosporidium* spp. infection in leopard geckos in Chiang Mai Provinces, Thailand.

Variables	Number of *Cryptosporidium* Positive (%)	Odd Ratios (95% CI)	*p* Value
Enclosure type (*n* = 23)			
Individual	7/18 (38.89)	1.17 (0.30–4.61)	0.826
Group	6/17 (32.29)	Ref.	
Sex (*n* = 35)			
Male	3/3 (100.00)	3.99 * (0.40–Inf) Ref.	0.249
Female	15/32 (47.88)		
Age (year) (*n* = 35)			
<1	2/4 (50.00)	0.94 (0.12–7.52)	0.952
1–5	16/31 (51.62)	Ref.	

Exact logistic regression analysis, * median unbiased estimates.

**Table 3 animals-16-00033-t003:** Univariable logistic regression analysis of variables associated with wasting syndrome in leopard geckos in Chiang Mai Provinces, Thailand.

Variables	Number of Wasting Positive (%)	Odd Ratios (95% CI)	*p* Value
*Cryptosporidium* nPCR (*n* = 35)			
Positive	6/18 (33.33)	8.00 (0.847–75.56)	0.070
Negative	1/17 (5.88)	Ref.	
Cage type (*n* = 35)			
Individual	2/13 (15.38)	Ref.	
Group	5/22 (18.18)	1.62 (0.27–9.85)	0.602
Sex (*n* = 35)			
Male	0/3 (0.00)	Ref.	
Female	7/32 (21.88)	1.00 * (0.00–10.30)	≅1.000
Age (year) (*n* = 35)			
<1	1/4 (25.00)	1.39 (0.12–15.81)	0.791
1–5	6/31 (19.35)	Ref.	
Body weight		0.89 (0.82–0.97)	0.007 **
Tail diameter		0.17 (0.04–0.78)	0.022 **
BCS		0.04 * (0–0.35)	0.001 **

Exact logistic regression analysis, * median unbiased estimates; ** Statistically Significant difference.

**Table 4 animals-16-00033-t004:** Multivariable logistic regression analysis of factors associated with wasting syndrome in leopard geckos in Chiang Mai, Thailand.

Variables	Odd Ratios (95% CI)	*p* Value
More than 1 gecko	13.64 (1.28–145.78)	0.031 *
<1 year	7.14 (0.60–84.71)	0.119
*Cryptosporidium* positive	11.15 (1.78–69.98)	0.010 *
Tail diameter (mm)	0.33 (0.21–0.50)	<0.001 *

* Statistically Significant difference.

## Data Availability

Additional data supporting the findings of this study are available from the corresponding authors upon reasonable request.

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
