# Peer review of "Cryptosporidium varanii Infection in Captive Leopard Gecko (Eublepharis macularius) and Its Association with Wasting Syndrome in Thailand"

_animals, 2025, doi:10.3390/ani16010033_

Round 1
Reviewer 1 Report
Comments and Suggestions for Authors
This study documents a high burden of C. varanii in a Thai leopard gecko farm and shows a significant association with wasting, supported by sequencing and a confirmatory necropsy. The longitudinal dataset and triangulation (nPCR-RFLP, sequencing, histology, acid-fast stain) are strengths. However, enclosure-pooled sampling, modest size, and week-wise statistics limit causal inference and effect precision; mixed-effects modeling and individual sampling would strengthen conclusions. Below are detailed comments and suggestions for improvement.
***Introduction***
-
Lines 56–66/67–79: Background is adequate; add a succinct differential for wasting (nutritional, parasites other than Crypto., bacterial enteritis) to reduce overattribution. Cite relevant reptile husbandry stressors already hinted at.
***Materials & Methods***
- Line 99: “Chaing Mai University” → “Chiang Mai University.” Ensure consistency in Ethics and Affiliations.
-
Sampling unit, Lines 108–115: Clarify that the analytical unit for infection prevalence is enclosure-week and enclosure across time; when reporting “individual prevalence,” explain the rule mapping pooled positives to individuals (e.g., any positive in enclosure → all residents counted positive?). Currently this can inflate individual prevalence. Consider a sensitivity analysis restricting “individual positive” to necropsied/PCR-confirmed individuals.
-
Wasting definition, Lines 117–125: BCS≤2 is reasonable; specify examiner blinding and intra-rater reliability if available.
-
Molecular identification, Lines 133–144 / 145–158: Methods are standard. Report negative controls per PCR batch and any contamination safeguards. Provide amplicon length in the first PCR as well. Deposit 12 sequences with clear sample IDs in a table (S3 lists accessions; ensure mapping sample↔accession in the main text).
-
Statistics, Lines 181–200/270–281: Replace week-by-week two-sample tests with linear mixed-effects models (random effects: enclosure and/or gecko ID; fixed effects: time, infection status; AR(1) correlation for repeated measures). This increases power and accounts for within-enclosure correlation. For infection–wasting association, present multivariable exact logistic regression (covariates: age, sex, cage type) rather than univariable only. The “GIGA Odds Ratio calculator” web link and an A/B testing blog in References are not suitable academic sources—please remove and cite standard statistical texts or software (e.g., R packages).
-
189–200 / Table 1: If retaining week-specific contrasts, control the family-wise error (Holm) or false-discovery rate across timepoints and outcomes (BW, tail, BCS).
***Results***
-
Lines 207–223: Add binomial exact CI for individual prevalence (already for enclosures). Include a small phylogeographic paragraph interpreting T→C polymorphism (Table S3) and whether clusters track host or geography.
-
Lines 224–226: Define “persistent” (e.g., ≥3 positive of 4 timepoints) and summarize counts per enclosure; Table S2 is useful—add a concise numeric summary in the main text.
-
Lines 270–281 / Table 1: Add effect sizes (mean differences with 95% CI) rather than p-values only; visualize trajectories with ribbons (mean±SE) and per-enclosure spaghetti to show heterogeneity.
***Discussion***
-
Lines 358–366 / 369–377: You appropriately caution that some infected geckos remained clinically well. Explicitly state that association ≠ causation given pooled sampling and confounding (season/breeding, temperature).
-
Lines 319–339: Consider rephrasing “revise description of C. varanii to gastrointestinal parasite” more cautiously—state that both gastric and intestinal involvement can occur in reptiles, with species/host differences.
-
Lines 378–393/394–413: Strengthen by proposing practical measures: quarantine newly purchased stock; enclosure-dedicated tools; effective disinfectants (cite studies already listed); and test-and-separate strategies. The references on disinfectant efficacy are well utilized; however, it may be beneficial to include information about peroxygen compounds that are used in herpetological facilities.
***Ethics / Reporting / Data***
-
Lines 441–447: Ethics statement present; correct “Chaing.”
-
Data availability (Lines 446–447): Prefer depositing anonymized dataset (weekly BW/tail/BCS per enclosure; nPCR results per time; sequences) in a public repository.
***Figures & Supplement***
-
Figure 4 legend (Lines 213–215): Spell “patterns.” Add enzyme source/catalog.
-
Table S1/S2/S3: Ensure titles are typo-free (“Nucelotide” → “Nucleotide”). Standardize “SSU rRNA gene (18S).”
Author Response
We thank the Reviewer#1 for their positive evaluation and helpful suggestions, which have substantially improved the clarity and rigor of our manuscript.
The reviewers’ comments are in Palatino Linotype font, followed by our responses in the highlighted text (in ‘Yellow’ of the comments by Reviewer#1; in ‘Bright Blue’ of the comments by Reviewer#2; and in ‘Green’ of the comments by Reviewer#3).

Reviewer 2 Report
Comments and Suggestions for Authors
This manuscript presents a very interesting and timely study. As exotic animals are increasingly kept as companion species, understanding their diseases is essential not only to safeguard their health and welfare but also to identify potential reservoir hosts and possible transmission routes of pathogens relevant to humans. From this perspective, your work brings valuable insight and contributes meaningfully to a broader One Health understanding of Cryptosporidium varanii infections in leopard geckos.
The study is well-designed, the methodology is solid, and the findings are clearly significant, especially given the scarcity of data from Thailand and Southeast Asia. The demonstrated association between infection and wasting syndrome adds clinical relevance, and the integrated approach (molecular analysis, histopathology, and field observations) strengthens the conclusions.
That said, a few aspects need clarification or expansion to improve the manuscript's clarity and overall impact.
Incomplete sentence in the Methods section: The sentence beginning with “At any time during the study, if an animal tested positive…” is incomplete. It should be finished—e.g., by stating that a necropsy was performed.
Figure 7 arrow: Please ensure that the arrow clearly points to the oocysts. At the moment, the indication is not entirely evident.
Number of necropsied animals: It is not clear how many animals were necropsied. The text suggests that only one animal underwent necropsy; this should be confirmed and explicitly stated.
Histopathology section needs more detail. Adding a bit more description would enrich the section: Were lesions present in the stomach?
Figure 8 placement: The figure looks slightly mispositioned on the page. Adjusting its layout would improve readability.
One Health perspective: This is an excellent opportunity to highlight the broader relevance of your findings. Expanding on this point, even briefly, would elevate the manuscript. For example:
This is a strong and meaningful study, with clear relevance for reptile health, breeders, veterinarians, and One Health research. With a few clarifications and minor improvements, the manuscript will be even clearer and more impactful.
Author Response
We thank the Reviewer#2 for their positive evaluation and helpful suggestions, which have substantially improved the clarity and rigor of our manuscript.
The reviewers’ comments are in Palatino Linotype font, followed by our responses in the highlighted text (in ‘Yellow’ of the comments by Reviewer#1; in ‘Bright Blue’ of the comments by Reviewer#2; and in ‘Green’ of the comments by Reviewer#3).

Reviewer 3 Report
Comments and Suggestions for Authors
L16-18 my suggestion is to change to “This study investigated whether Cryptosporidium infection contributes to the development of wasting syndrome in captive leopard geckos from a commercial breeding facility in Chiang Mai, Thailand.”
L22-23 change to “Slightly over half of the geckos and enclosures were infected with Cryptosporidium, while one in five individuals presented with wasting syndrome.”
L24 at the beginning of the sentence please use full genus name, check it throughput the manuscript
From the simple summary it is not clear how exact species of the genus Cryptosporidium was identified
L31-32 Italicized have to be genus name
L33-35 and L19-L21 is very similar sentences, in simple summary even more simplified description of the methods should be provided
L35-36 abbreviations must be spelled out BW and BCS
L70-74 so which species C. serpentis or C. varanii is more hazardous for geckos? It can be disclosed in the text
L90 “were obtained from”
L91 were instead of are
Figure 2: please use either lower- or uppercase consistently in both the figure caption and the figure itself
L127 scores
L130-132 I have two questions considering this section. The answers have to be incorporated in the text. 1) what volume of buffer/water was used in the final step of DNA purification? 2) Was the DNA concentration measured, and if so, was it diluted and to what concentration?
L137 really PCR without final extension at 72 degree?
L134-140 approximate length of PCR products in silico expected after first and the second round of nested PGRs have to be presented here.
L141-144 I am not familiar with Crystosporidium molecular identification, therefore for me it is not clear why to use the RFLP with only one restriction nuclease? Authors should describe it in the text
L163 “as previously topic” correct English
L167-170 correct English, now it is not grammatically clear
L185-186 change to “The overall proportion of wasting syndrome in the breeding farm was calculated as the percentage of geckos with a BCS ≤ 2 among all geckos examined (n = 35).”
L202-203 the title should be simplified
L205 change SSU to 18S rRNA gene, here and throughout the manuscript; it is confusing when these are used as synonyms.
Tables S1-S3 must be presented consecutively in the text, first Table S1, then Table S2, and finally Table S3
L204-223 not clear why restriction analysis is here needed. The length of sequences should be indicated. Based on sequences provided in the Supplementary table they are 734 bp long, so in the light of this result it appears that 833 bp band was sequenced, and why to do then restriction analysis with only one restriction endonuclease? I do not see the practical significance of such analysis.
The range of BLAST values (for instance 92.5-95.0% similarity) compared to Cryptosporidium serpentis has to be included.
Figure 5 is inappropriate, to resolve phylogenetic relationships the rooted analysis is needed; čtherefore please redo phylogenetic analysis
L233 correct, full genus name is needed also please write Latin name of the species in italic
In methodological part is written “with branch reliability assessed by 1000 bootstrap replicates”; however bootstrap values are not presented in the resulted phylogenetic tree; therefore, we cannot assess the reliability of the branching, now it is not clear to which species the identified one is most closely related. Furthermore, there is no appropriate description of phylogenetic results.
L239-269 too detailed division 3.3.1, 3.3.2, 3.3.3, 3.3.4; now there are only a few lines in the subsections, which shows that this division is not correct
L296 yes, authors has observed that Cryptosporidium infection was statistically associated with wasting syndrome. However, p value is quite large (0.023). How can the authors explain this observation?
L312 “For specie” please correct
L312-313 “C. parvum were reported in leopard geckos“ this should be also revealed in Introduction, as in this part of the introduction only two Cryptosporidium species have been mentioned
L314 it has to be – 18S
L313-315 this is not discussion, but repetition of results
L415 “patients” this word is suits for humans and should be changed
In the Discussion, the authors should elaborate on how Cryptosporidium varanii infection in captive leopard geckos could be linked to humans and may represent a potential threat to human health.
Author Response
We thank the Reviewer#3 for their positive evaluation and helpful suggestions, which have substantially improved the clarity and rigor of our manuscript.
The reviewers’ comments are in Palatino Linotype font, followed by our responses in the highlighted text (in ‘Yellow’ of the comments by Reviewer#1; in ‘Bright Blue’ of the comments by Reviewer#2; and in ‘Green’ of the comments by Reviewer#3).

Round 2
Reviewer 1 Report
Comments and Suggestions for Authors
Accept in present form
Author Response
Thank you very much for taking the time to review this revise version of manuscript and accepted in present form.
Reviewer 3 Report
Comments and Suggestions for Authors
My major concerns were considering the need of RFLP analysis and unrooted phylogenetic tree; authors in general addressed all of my comments. Please find two clarifications needed below
1.
“L141-144 I am not familiar with Crystosporidium molecular identification, therefore for me it is not clear why to use the RFLP with only one restriction nuclease? Authors should describe it in the text
Response:
We used only SspI for RFLP analysis because this enzyme generates a characteristic species-specific restriction pattern that distinguishes C. varanii from closely related Cryptosporidium spp., such as C. serpentis, C. parvum, as demonstrated in previous studies [Xiao et al., 1999].
Ref: Xiao, L., Morgan, U. M., Limor, J., Escalante, A., Arrowood, M., Shulaw, W., Thompson, R. C., Fayer, R., & Lal, A. A. (1999). Genetic diversity within Cryptosporidium parvum and related Cryptosporidium species. Applied and environmental microbiology, 65(8), 3386–3391. https://doi.org/10.1128/AEM.65.8.3386-3391.1999”
Now I can understand the need of RFLP, however, please add more clarification in methodological part L178-179. The authors may include a sentence similar to the one above in their response to my comment.
2.
L269-270 “When compared with serpentis, BLAST analysis revealed 92.70%-99.86% identity” this seems vague for me if the sequence similarity is 99.86% it should represent the same species, so C. varanii cannot be discriminated from C. serpentis, but based on phylogenetic results opposite conclusions should be drawn.
Author Response
We thank the Reviewer#3 for their positive evaluation and helpful suggestions, which have substantially improved the clarity of our manuscript.
Comments and Suggestions for Authors
My major concerns were considering the need of RFLP analysis and unrooted phylogenetic tree; authors in general addressed all of my comments. Please find two clarifications needed below
- “L141-144 I am not familiar with Crystosporidium molecular identification, therefore for me it is not clear why to use the RFLP with only one restriction nuclease? Authors should describe it in the text
Response:
We used only SspI for RFLP analysis because this enzyme generates a characteristic species-specific restriction pattern that distinguishes C. varanii from closely related Cryptosporidium spp., such as C. serpentis, C. parvum, as demonstrated in previous studies [Xiao et al., 1999].
Now I can understand the need of RFLP, however, please add more clarification in methodological part L178-179. The authors may include a sentence similar to the one above in their response to my comment.
Response:
We added a sentence to clarify the relative impact based on current evidence.
Changes in manuscript: Line 179-182
“The restriction endonuclease SspI was employed for RFLP analysis, as this enzyme produces a characteristic restriction profile enabling discrimination of C. varanii from closely related Cryptosporidium spp., including C. serpentis and C. parvum, as reported previously [14,15].”
- L269-270 “When compared withserpentis, BLAST analysis revealed 92.70%-99.86% identity” this seems vague for me if the sequence similarity is 99.86% it should represent the same species, so C. varanii cannot be discriminated from C. serpentis, but based on phylogenetic results opposite conclusions should be drawn.
Response:
We agree that high BLAST identity alone may appear ambiguous. We have revised the manuscript to clarify that species identification was based on phylogenetic analysis rather than BLAST similarity alone.
Changes in manuscript: Discussion Line 406-411
“The observation of 99.86% identity with C. serpentis likely reflects the conserved nature of the target 18s rRNA gene and highlights a known limitation of BLAST-based species assignment for Cryptosporidium. Consequently, species designation in this study was based primarily on phylogenetic inference, which demonstrated robust clustering with reference C. varanii sequences and separation from C. serpentis, consistent with previously published classifications.”